# Comparison of Retinal Structural and Neurovascular Changes between Patients with and without Amyloid Pathology

**DOI:** 10.3390/jcm12041310

**Published:** 2023-02-07

**Authors:** Sangwoo Moon, Sumin Jeon, Sook Kyeong Seo, Da Eun Kim, Na-Yeon Jung, Seung Joo Kim, Myung Jun Lee, Jiwoong Lee, Eun-Joo Kim

**Affiliations:** 1Department of Ophthalmology, Pusan National University Hospital, Pusan National University School of Medicine and Biomedical Research Institute, Busan 49241, Republic of Korea; 2Department of Neurology, Pusan National University Hospital, Pusan National University School of Medicine and Biomedical Research Institute, Busan 49241, Republic of Korea; 3Department of Neurology, Pusan National University Yangsan Hospital, Pusan National University School of Medicine, Yangsan 50612, Republic of Korea; 4Department of Neurology, Gyeongsang National University School of Medicine and Gyeongsang National University Changwon Hospital, Changwon 51472, Republic of Korea

**Keywords:** Alzheimer’s disease, Aβ pathology, optical coherence tomographic angiography

## Abstract

To evaluate whether an impaired anterior visual pathway (retinal structures with microvasculature) is associated with underlying beta-amyloid (Aβ) pathologies in patients with Alzheimer’s disease dementia (ADD) and mild cognitive impairment (MCI), we compared retinal structural and vascular factors in each subgroup with positive or negative amyloid biomarkers. Twenty-seven patients with dementia, thirty-five with MCI, and nine with cognitively unimpaired (CU) controls were consecutively recruited. All participants were divided into positive Aβ (A+) or negative Aβ (A−) pathology based on amyloid positron emission tomography or cerebrospinal fluid Aβ. The retinal circumpapillary retinal nerve fiber layer thickness (cpRNFLT), macular ganglion cell/inner plexiform layer thickness (mGC/IPLT), and microcirculation of the macular superficial capillary plexus were measured using optical coherence tomography (OCT) and OCT angiography. One eye of each participant was included in the analysis. Retinal structural and vascular factors significantly decreased in the following order: dementia < MCI < CU controls. The A+ group had significantly lower microcirculation in the para- and peri-foveal temporal regions than did the A−. However, the structural and vascular parameters did not differ between the A+ and A− with dementia. The cpRNFLT was unexpectedly greater in the A+ than in the A− with MCI. mGC/IPLT was lower in the A+ CU than in the A− CU. Our findings suggest that retinal structural changes may occur in the preclinical and early stages of dementia but are not highly specific to AD pathophysiology. In contrast, decreased temporal macula microcirculation may be used as a biomarker for the underlying Aβ pathology.

## 1. Introduction

Alzheimer’s disease dementia (ADD) is the most common neurodegenerative dementia in the elderly and is associated with the accumulation of beta-amyloid (Aβ) plaques and tau neurofibrillary tangles [1]. Diagnostic biomarkers reflecting the underlying amyloid or tau pathology of Alzheimer’s disease (AD) have been actively developed. Of these, amyloid and tau positron emission tomography (PET) imaging and cerebrospinal fluid (CSF) Aβ1-42 or phosphorylated tau are representative biomarkers for ADD [2]. However, these tend to be expensive, invasive, or available only in tertiary hospitals or specialized laboratories. Alternative biomarkers that are inexpensive, less invasive, and readily available are needed.

Patients with ADD commonly show visuospatial dysfunction that is mainly attributed to damage of the parieto-occipital or temporo-occipital visual pathways [3]. Ocular manifestations of patients with ADD, including retinal ganglion cell (RGC) loss and retinal nerve fiber layer (RNFL) thinning, have also been reported as possible biomarkers in recent studies using optical coherence tomography (OCT), a useful tool that measures circumpapillary RNFL thickness (cpRNFLT) and macular ganglion cell/inner plexiform layer thickness (mGC/IPLT). However, results for regional patterns of cpRNFLT and mGC/IPLT reduction among studies have been inconsistent [4,5,6,7].

Cerebrovascular dysfunction in association with vascular deposition of Aβ is one of the major pathophysiological mechanisms of AD [8]. The brain and retina have similar anatomic and physiologic vascular traits because they emerge from the same embryological origin, the forebrain [9,10]. Therefore, the cerebral vasculature may be screened by assessing the retinal vasculature using retinal digital imaging techniques.

Several studies using laser Doppler ultrasonography or a retinal function imager (RFI) have revealed narrowed central retinal vessels with a decreased blood flow rate in patients with mild cognitive impairment (MCI) and ADD [11,12,13]. Additionally, post-mortem studies have shown that Aβ accumulates inside or around RGCs and retinal vessels in patients with ADD [14,15]. The retina and its microvascular network are very complicated; therefore, it would be difficult to identify early pathological alterations in the retinal microvasculature by only measuring the blood flow of larger vessels on RFI or Doppler ultrasonography [16].

On the other hand, OCT angiography (OCTA) can be used to non-invasively visualize the retinal microvascular network involving multiple retinal layers in a reproducible manner without a contrast agent [17]. Prior studies using OCTA have reported an enlarged foveal avascular zone (FAZ) and reduced retinal vascular density in patients with ADD or MCI compared to that in controls [18,19,20,21], while some studies have shown no differences in the area of the FAZ or the retinal vessel density between them [22,23]. The reason for this discrepancy is not entirely clear. However, most previous studies have investigated the pregeniculate visual pathway in patients with clinically diagnosed ADD. This means that patients with non-AD pathology might be included, and thus, they might not demonstrate the effects of Aβ pathology on the retina and its vasculature [18,19,21]. Only a few studies have examined degenerative changes in the retina and its microvasculature in patients with ADD with underlying Aβ pathology [20,22,23].

In this study, we first analyzed the structural changes in the mGC/IPLT and cpRNFLT with microvasculature in patients with clinically diagnosed dementia, MCI, and in cognitively unimpaired (CU) controls using OCT and OCTA, and we investigated the diagnostic performance (cut-off value) of OCT/OCTA variables discriminating dementia and/or MCI group from the CU group. Second, to evaluate whether the impaired anterior visual pathway (retinal structures with microvasculature) is truly associated with the underlying amyloid pathology, we further compared the OCT and OCTA parameters in each subgroup with positive or negative amyloid biomarkers and examined their usefulness as biomarkers for predicting the underlying amyloid pathology.

## 2. Materials and Methods

### 2.1. Study Design and Population

We recruited 234 eyes of 117 participants with clinically diagnosed dementia, MCI, or CU at the Neurological Clinic of Pusan National University Hospital from September 2019 to July 2021. All participants were evaluated by a neurologist (E.J.K.) with expertise in neurodegenerative diseases who underwent clinical interviews, neurological examinations, and detailed neuropsychological testing [24]. Patients with dementia and MCI fulfilled the National Institute on Aging—Alzheimer’s Association (NIA-AA) core clinical criteria for all-cause dementia and MCI, respectively [25,26]. CU participants, who had cognition within normal limits, were also enrolled as controls. All participants underwent amyloid PET or cerebrospinal fluid (CSF) analyses for AD biomarkers. Amyloid PET was visually assessed by a nuclear physician who was blinded to clinical data, using a previously described method [27]. CSF Aβ1-42 was assessed using detailed methods and cut-off values described elsewhere [28]. The participants were divided into Aβ-positive (A+) and Aβ-negative (A−) groups. A+ refers to Aβ pathology (CSF Aβ1-42 < 631.8 pg/mL or positive deposits on amyloid PET by visual inspection) and A− to non-Aβ pathology (CSF Aβ1-42 ≥ 631.8 pg/mL or negative deposits on amyloid PET by visual inspection).

Thorough ophthalmologic examinations, including best-corrected visual acuity (BCVA), slit lamp examination, intraocular pressure (IOP) measurement using Goldmann applanation tonometry, gonioscopy, dilated fundus examination, stereoscopic optic disc, and RNFL photography (AFC-210; Nidek, Aichi, Japan), biometry using IOLMaster (Carl Zeiss Meditec, Dublin, CA, USA), standard automated perimetry (Carl Zeiss Meditec, Dublin, CA, USA), ultrasonic pachymetry (Pachmate; DGH Technology, Exton, PA, USA), and keratometry (Auto Kerato-Refractometer, ARK-510A; NIDEK, Hiroshi, Japan) were conducted in all participants.

The exclusion criteria were glaucoma, macular degeneration, retinal vascular disease, including diabetic retinopathy and retinal vein occlusion, corneal or ocular media opacity, history of ocular trauma, astigmatism ≥ 3.0 diopter, spherical equivalent ≥ 6.0 diopter, BCVA < 20/40, and any ocular surgery except for uncomplicated cataract extraction. Additional medical exclusion criteria included uncontrolled hypertension that caused vascular eye changes [29].

This study was approved by the institutional review board of the Pusan National University Hospital (approval no. 1908-004-081) and registered at ClinicalTrials.gov (approval no. NCT05475158). Written informed consent was obtained from all participants enrolled in the study.

### 2.2. OCT Imaging

A Zeiss Cirrus Spectral domain (SD)-OCT 5000 with AngioPlex™ (Carl Zeiss Meditec, Dublin, CA, USA, software version 6.0), at a scan rate of 68,000 A-scans per second and a central wavelength of 840 nm, was used to measure the mGC/IPLT and cpRNFLT. After pupil dilatation, the macular cube 200 × 200 scan protocol (6 × 6 mm^2^) and the optic disc cube 200 × 200 scan protocol (6 × 6 mm^2^) were performed for each eye. The software algorithm automatically identified the outer boundaries of the RNFL and IPL. The mGC/IPLT, within a 14.13 mm^2^ elliptical annulus area centered on the fovea, was calculated as the distance between these two boundaries. The average, minimum, and six sectoral (superotemporal, superior, superonasal, inferonasal, inferior, and inferotemporal) mGC/IPLT values were measured (Figure 1A). The optic disc cube protocol was used to measure the cpRNFLT and to generate a cpRNFLT map. A 3.46 mm-diameter circle scan consisting of 256 A-scans was located around the optic disc. The average, four sectoral (temporal, superior, nasal, and inferior), and twelve clock-hour cpRNFLT were measured (Figure 1B).

### 2.3. OCTA Imaging

ZEISS Angioplex™ OCT angiography has tracking software known as FastTrac^TM^ retinal-tracking technology that can be used to reduce motion artifacts [17]. The OCT microangiography-complex algorithm was used to analyze the change in complex signals (both the phase and intensity information contained within sequential B-scans performed at the same position) and generated en face microvascular images [17]. In macular angiography, we focused on the superficial capillary plexus (SCP) located in the RNFL and ganglion cell layer, which plays a major role in the metabolic demand of macular ganglion cells [30]. The SCP consists of the capillaries in the area between the internal limiting membrane and the IPL. The average vessel density (VD) and perfusion density (PD) of the SCP were automatically measured on the Early Treatment of Diabetic Retinopathy Study (ETDRS) grid [31]. Three concentric circles of the macular SCP are represented in Figure 1C,D, including a central, inner circular zone (1.0–3.0 mm from the fovea, referred to as the parafoveal region), and an outer circular zone (3.0–6.0 mm from the fovea, referred to as the perifoveal region) [16,31]. VD was defined as the total length and PD as the total area of perfused vasculature per unit area in each subfield. The area, perimeter, and circularity index (4πA/P^2^, where A is the area and P is the perimeter) of the FAZ were measured. A circularity index closer to 0 indicates an irregular shape, and a value closer to 1 indicates a circular shape [32].

All OCT and OCTA scans were performed by an experienced examiner, and all scans were reviewed individually by two investigators (S.M. and J.L.) for quality evaluation (segmentation errors and motion artifacts). Substandard scans or image scans with a signal strength of <7 were excluded.

### 2.4. Statistical Analysis

Data distribution normality was checked using the Kolmogorov–Smirnov test. One-way analysis of variance (one-way ANOVA) and the Kruskal–Wallis test with post-hoc correction for multiple comparisons were used to analyze differences in demographic and ophthalmological characteristics in continuous variables among the clinical diagnostic groups. To analyze differences in categorical variables, we used the chi-square test or Fisher’s exact test. The Student’s t-test or Mann–Whitney U test for continuous variables was used for comparison between the A+ and A− groups after adjustment for the Mini-Mental Status Examination (MMSE). The correlation between OCT or OCTA parameters and MMSE scores was estimated using Pearson or Spearman’s rank correlation coefficients.

Receiver operating characteristic (ROC) curves were drawn by plotting sensitivity against 1-specificity, and areas under the ROC curves (AUCs) were used to evaluate the diagnostic performance of the OCT/OCTA parameters. Ideal cut-off points were derived using the Youden index [33]. Binary logistic regression analyses were also performed to explore the predictors of OCT/OCTA parameters discriminating dementia and/or MCI from CU after adjusting for age and sex or discriminating A+ from A− groups after adjusting for age, sex, and MMSE. Only a single OCT/OCTA variable was included at a time in all statistical models.

All statistical analyses were performed using the language R (http://cran.r-project.org accessed on from September 2021 to December 2022), version 4.0.5, and SPSS software, version 22.0 (IBM Corp., Armonk, NY, USA). In all analyses, a *p*-value < 0.05 was considered statistically significant.

## 3. Results

### 3.1. Demographics and Clinical Characteristics

Among the 234 eyes of the 117 participants, 120 eyes were excluded during screening (32 for glaucoma, 29 for macular degeneration, 15 for diabetic retinopathy or retinal vascular disease, 8 for ocular media opacity, 6 for optic disc abnormalities, 2 for intraocular surgeries within 12 months, 2 for visual field abnormalities, two for laser therapies, 1 for BCVA < 20/40, and 23 for OCT or OCTA measurement errors, such as motion artifacts, vitreous floaters, signal strength, or segmentation errors). One eye of each participant was included in the analysis. In cases where both eyes met the eligible criteria, the eye with the worse mean values of structural and vascular parameters on OCT/OCTA images was selected [34]. When the mean values of the parameters were similar in both eyes, the eyes with better-quality OCT/OCTA images were included. Finally, 71 eyes of 71 living participants were eligible for the study and classified into 3 clinical diagnostic groups: 27 eyes of patients with dementia, 35 eyes of patients with MCI, and 9 eyes of CU controls. The 27 patients with dementia comprised 26 with clinically diagnosed ADD and 1 with subcortical ischemic vascular dementia. All the participants underwent amyloid PET (*n* = 69), CSF studies for Aβ analysis (*n* = 11), or both (*n* = 9).

Based on amyloid PET or CSF Aβ level results, all participants were subdivided into A+ (*n* = 42 eyes) or A− (*n* = 29 eyes) groups. The clinical diagnostic groups were classified as A+ or A− as follows: 22 eyes (A+) and 5 eyes (A−) of patients with dementia, 17 eyes (A+) and 18 eyes (A−) of patients with MCI, and 3 eyes (A+) and 6 eyes (A−) of CU controls. The mean interval between the OCT/OCTA test and the amyloid PET scan or CSF study for Aβ_1-42_ was 1.1 ± 1.0 years. The proportions of eyes (number) from A+ participants in the dementia, MCI, and CU control groups were 81.5%, 48.6%, and 33.3%, respectively. These differences differed significantly among the clinical groups (*p* = 0.008).

The demographic and clinical characteristics of the participants are summarized in Table 1. The dementia group had significantly worse general cognitive indices (MMSE and Clinical Dementia Rating) and a higher proportion of Aβ positivity than did the MCI and CU groups. The A+ group also showed significantly worse general cognitive indices than the A− group. There were no significant differences in other variables, including age, sex, blood pressure, vascular risk factors, BCVA, IOP, spherical equivalent, central corneal thickness, and axial length, between the clinical diagnostic groups, or the A+ and A− groups. All patients with A− and dementia were male. CU controls with A+ had significantly worse performance on the MMSE and higher systolic blood pressure than those with A−. The demographic and clinical characteristics did not significantly differ between patients with A+ and those with A− in the MCI group (Appendix A).

### 3.2. Comparison of OCT and OCTA Parameters and Diagnostic Accuracy among Clinical Diagnostic Groups

The OCT and OCTA parameters significantly differed among the clinical diagnostic groups (Table 2). Regarding retinal structural factors, the superior quadrant, and 5 and 6 clock-hour cpRNFLTs, were significantly different between the 3 groups, with generally smaller thickness in the direction of dementia < MCI < CU controls. Post-hoc analysis revealed that all these retinal structural factors in the dementia group were significantly lower than those in the CU group. Regarding retinal vascular factors, the dementia and MCI groups had significantly lower FAZ circularity indices than the CU group. The parafoveal vascular parameters were not significantly different among the three groups, whereas the perifoveal temporal and mean vascular parameters had significantly lower VDs and PDs in the following order: dementia < MCI < CU controls. Significant differences of VDs and PDs between the dementia and CU control groups remained after the post-hoc analysis. The perifoveal superior PD was also significantly lower in the dementia group than in the CU group after post-hoc analysis. However, these results were no longer significant after correction for multiple testing (Appendix A).

Table 3 summarizes the diagnostic accuracies of OCT and OCTA parameters. To discriminate patients with dementia and MCI from CU controls, the FAZ circularity index (AUC, 0.794; sensitivity, 66.1%; specificity, 88.9%) and the superior quadrant cpRNFLT (AUC, 0.772; sensitivity, 75.8%; specificity, 77.8%) were the most accurate, with cut-off points of 0.68 and 123.0 μm, respectively. To distinguish dementia from CU, the FAZ circularity index (AUC, 0.844; sensitivity, 77.8%; specificity, 88.9%) and the superior quadrant cpRNFLT (AUC, 0.821; sensitivity, 66.7%; specificity, 88.9%) were more accurate than other parameters, with cut-off points of 0.68 and 118.0 μm, respectively. To differentiate MCI from CU, the AUC of the FAZ circularity index was 0.756 (sensitivity, 71.4%; specificity, 77.8%), with a cut-off point of 0.71. These parameters were also significant predictors in binary logistic regression analyses, although significant differences no longer existed after correction for multiple testing (Appendix A).

### 3.3. Comparison of OCT and OCTA Parameters and Diagnostic Accuracy According to Aβ Positivity

Significantly different parameters according to Aβ positivity after adjusting for MMSE scores are shown in Table 4. The retinal structural parameters did not differ between the A+ and A− groups. Among the retinal vascular factors, para- and peri-foveal temporal VDs and PDs were significantly lower in the A+ group than in the A− group. In the dementia group, no significant structural or vascular differences were observed between the A+ and A− groups. In the MCI group, the nine clock-hour and temporal quadrant cpRNFLT values were significantly increased in the A+ group compared to the A− group, while para- and peri-foveal temporal VDs and PDs were decreased in the A+ group. However, these results were no longer significant after correction for multiple testing (Appendix A). In the CU group, no significant structural or vascular differences were observed between the A+ and A− groups after adjusting for MMSE scores.

In all participants, perifoveal temporal VD (AUC, 0.688; sensitivity, 85.7%; specificity, 55.2%), PD (AUC, 0.684; sensitivity, 85.7%; specificity, 58.6%), and parafoveal temporal PD (AUC, 0.683; sensitivity, 64.3%; specificity, 72.4%) were more accurate than other parameters in differentiating the A+ from A− groups, with cut-off points of 16.88 mm/mm^2^, 0.41, and 0.40, respectively. In the MCI group, the parafoveal temporal PD (AUC, 0.742; sensitivity, 82.4%; specificity, 66.7%), 9 clock-hour cpRNFLT (AUC, 0.727; sensitivity, 70.6%; specificity, 77.8%), and perifoveal temporal VD (AUC, 0.725; sensitivity, 88.2%; specificity, 50.0%) parameters were more accurate than the others in discriminating underlying Aβ from non-Aβ pathology, with cut-off points of 0.41, 54.0 μm, and 16.88 mm/mm^2^, respectively. In the CU group, the structural factors, superotemporal, inferior, and inferotemporal mGC/IPLT, vascular factors, FAZ circularity index, and para- and peri-foveal temporal VD and PD showed accurate diagnostic performance in differentiating the A+ from A− groups (0.775 ≤ AUCs ≤ 0.988) (Appendix A). The parameters with the best performance in discriminating between A+ and A− in the whole group were also significant predictors in binary logistic regression analyses, although the significance no longer existed after correction for multiple testing (Appendix A).

Appendix A show the demographic and clinical characteristics and OCT/OCTA parameters of the A+ dementia, A+ MCI, and A− CU groups. Cognitive indices were significantly different among the three groups, with worse performance in the direction of A+ dementia < A+ MCI < A− CU. Post-hoc analysis revealed that the A+ dementia and A+ MCI groups had significantly lower FAZ circularity indices and perifoveal temporal VD and PD than the A− CU group. In addition, parafoveal inferior VD, perifoveal mean PD, and total mean PD in the A+ dementia group were significantly lower than those in the A− CU group. However, again, these results were no longer significant after correction for multiple testing.

Figure 2 shows representative cases with OCTA images of A+ (A, C, E) or A− (B, D, F) MCI participants. The A+ MCI participant was a 67-year-old female with an MMSE score of 21. The A− MCI participant was a 66-year-old female with an MMSE score of 21. The perifoveal temporal vessel density (C, D) and perfusion density (E, F) were lower in the A+ MCI participant (VD, 14.7 mm/mm^2^; PD, 36.3%) than in the A− MCI participant (VD, 16.6 mm/mm^2^; PD, 41.2%).

### 3.4. Correlations between MMSE Score and OCT/OCTA Parameters

The FAZ circularity index (r = 0.280, *p* = 0.018) was significantly correlated with MMSE scores among all participants.

Whereas structural (12 clock-hour cpRNFLT (ρ = 0.380, *p* = 0.042), 3 clock-hour cpRNFLT (ρ = 0.374, *p* = 0.046)) and vascular (FAZ circularity index (ρ = 0.378, *p* = 0.043)) parameters were positively correlated with MMSE scores in the entire A− group, no significant structural or vascular parameters were correlated with MMSE scores in the entire A+ group.

## 4. Discussion

The main findings of this study were as follows: (1) Among the clinical diagnostic groups, as the severity of neurodegeneration increased, the structural and vascular parameters of OCT/OCTA tended to significantly decrease. The superior quadrant cpRNFLT (as a structural factor) and the FAZ circularity index (as a vascular factor) showed the best performance in differentiating dementia and/or MCI from CU controls. (2) The A+ group had significantly decreased microcirculation in the para- and peri-foveal temporal regions compared to those of the A− group. (3) In the dementia group, there were no significant differences in the structural or vascular parameters on OCT/OCTA according to Aβ positivity. However, in the MCI group, the A+ group had a greater temporal quadrant and 9 clock-hour cpRNFLT and lower para- and peri-foveal temporal VD and PD than those of the A− group. (4) In the CU group, the average, temporal, and inferior mGC/IPLT and FAZ circularity decreased more in the A+ group than in the A− group. (5) FAZ circularity was correlated with MMSE scores in all participants. However, no such correlation was observed in the A+ group.

The results of the present study, which showed a reduced superior quadrant and inferior cpRNFLT region in patients with dementia and/or MCI compared to those of CU controls, partially correspond to those of previous studies conducted in patients with ADD [4,5,6,35,36]. Cheung et al. [4] reported that all six sectors of the mGC/IPLT and superior quadrant cpRNFLT were reduced in patients with ADD compared to controls. Cunha et al. [5] showed that the average, superior, and inferior quadrant cpRNFLT, average mGC/IPLT, and macular full thickness decreased in patients with ADD compared to controls. Kesler et al. [6] also reported that the inferior and/or superior quadrant cpRNFLT was reduced in patients with ADD and MCI compared to controls. These studies suggest that retinal structural changes reflect neurodegenerative changes in AD pathophysiology [4,5,6,35,36]. However, the inclusion of patients with ADD defined only clinically in these studies may limit relevant interpretations regarding the underlying AD pathophysiology. In other words, patients with non-AD pathology may have been included in their studies as patients with ADD. Indeed, the dementia group in our study mostly consisted of patients clinically diagnosed with ADD (26 out of 27, 96%), but 4 of these participants (4 out of 26, 15%) turned out to have non-Aβ pathology.

Regarding the retinal microvasculature, in the present study, the FAZ circularity index and perifoveal temporal and mean vascular parameters (VDs and PDs) were reduced in patients with dementia or MCI compared to those of CU controls. This is also partly consistent with the results of previous studies that reported FAZ enlargement or reduced macular microcirculation in patients with clinically diagnosed ADD compared to controls [16,18,19,37]. Cerebrovascular impairment in ADD is related to the impairment of Aβ clearance, reduced angiogenesis due to Aβ, and direct Aβ deposits inside the walls of blood vessels [38,39]. Since the retina has embryological, anatomical, and physiological features that are similar to those of the brain, microvascular damage of the retina might be explained by the same pathophysiological mechanism as a cerebrovascular disease in patients with ADD [9,10]. However, the diagnosis of ADD in most prior OCTA studies was based on clinical criteria without confirming underlying amyloid pathology. Thus, interpretation of the actual effect of Aβ pathology on the microvasculature of the retina might also be limited [16,18,19,37].

Several recent studies conducted on patients with or without Aβ pathology have shown inconsistent results [20,22,23,40]. Haan et al. [22] reported no differences in the retinal vasculature between A+ ADD and A− CU controls. Kreeke et al. [23] demonstrated significantly higher retinal VD in preclinical A+ than in A− participants, with no difference in FAZ size between them. They explained that an early inflammatory reaction from hypoxia induces an increase in retinal blood flow in preclinical A+ patients. In contrast, another preclinical AD study showed inner foveal thinning and FAZ enlargement in A+ preclinical patients [20]. These discrepancies among studies justify the need for a study such as ours to examine the influence of Aβ pathology on OCT/OCTA parameters within each clinical diagnostic group.

In our study, although significantly reduced vascular parameters were detected in the whole A+ versus A− group, no differences in structural parameters were found. In the dementia group, no differences in OCT/OCTA parameters were observed between the A+ and A− groups. In contrast, the MCI and CU groups had significantly different parameters between the A+ and A− groups. Each A+ group may have underlying Aβ pathology, but the underlying pathology in each A− group may be speculated differently. While the A− dementia group would have various underlying pathologies other than Aβ, A− MCI or A− CU participants seemed to have a very low risk of progression to neurodegenerative disease [28]. Therefore, our data may be interpreted to imply that contributions of Aβ pathology to retinal degeneration and microvascular changes begin at the preclinical stage (CU group) and play a role in the early and mild stages of disease (MCI group) but may not be emphasized in the dementia stage. Similar retinal structural changes have been reported in other neurodegenerative diseases, such as frontotemporal dementia, Parkinson’s disease, and Huntington’s disease [41,42,43]. These findings support the hypothesis that previous OCT/OCTA data, thinning of the retina, and reduced retinal microcirculation in clinical patients with ADD do not reflect AD-specific neuronal degeneration of the retina but nonspecific neurodegenerative changes in heterogeneous neurodegenerative diseases. The positive associations between the MMSE score and OCT/OCTA parameters observed in the whole group or in the A− group were not found in the A+ group, which may also indicate that retinal degeneration with vascular changes is parallel to non-specific neurodegenerative progression.

The unexpectedly increased nine clock-hour and temporal cpRNFLT in the A+ MCI group may be explained by a paradoxical increase in RNFLT in the MCI stage, attributed to gliosis occurring prior to neuronal loss [44]. Approximately 50% of the retinal ganglion cells are located within 4.5 mm (16°) of the fovea center, a macula region that comprises only 7.3% of the total area [45]. In addition, all axon of retinal ganglion cells from the superior macula and some axons of the inferior macula are projected to the temporal quadrant of the disc [46]. Therefore, the contribution of Aβ pathology to retinal degeneration, gliosis, or inflammatory reaction, which can cause a paradoxical increase in RNFL thickness, might be more prominent at the nine clock-hour and temporal quadrant of the disc in the early/mild stage of the disease. However, it remains unclear whether this paradoxical increase in cpRNFLT occurs as an early phenomenon of neurodegeneration due to Aβ pathology alone. Further longitudinal and histopathological studies on retinal structural changes in various neurodegenerative diseases are required.

In contrast to the heterogeneous structural changes in the retina, a decrease in the para- or peri-foveal temporal microcirculation was consistently observed in the A+ group. Magnocellular neurons (M cells), the largest class of RGCs, are known to be primarily affected in AD, and the number of M cells is greatest in the temporal macula [47,48]. Therefore, in this study, the decrease in para- and peri-foveal temporal microcirculation in the A+ group is reasonable in terms of the anatomy and pathophysiology of AD.

Lastly, we detected a reduction in the FAZ circularity index and peri/parafoveal temporal VD and PD between the A+ dementia and A− CU groups, which was incompatible with the results of Hann et al.’s study [22]; however, this should be cautiously interpreted because of the small A− CU group (*n* = 6).

A major limitation of our study is the small sample size in each clinical group. No significant differences of OCT and OCTA parameters among clinical groups or between A + and A− groups were observed after correction for multiple testing, which might be due to the low statistical power caused by multiple variables considering the small and imbalanced sample sizes between the groups. For such a reason, we also acknowledge that the OCT and OCTA parameters with the best discriminating performances from the ROC curve were no longer significant in the binary logistic regression model after correction for multiple testing.

## 5. Conclusions

This is the first study to investigate the role of Aβ pathology in each clinical diagnostic group: dementia, MCI, and CU controls. Significant reductions in cpRNFLT and the FAZ circularity index were found to be associated with neurodegenerative processes, but not with or without Aβ pathology. Therefore, loss of the superior quadrant cpRNFLT and a decreased FAZ circularity index may be considered the most consistent phenomena in various neurodegenerative diseases, and are not highly specific to AD. In contrast, decreased para- and peri-foveal temporal microcirculation might be a good biomarker for predicting the underlying Aβ pathology. Longitudinal studies with larger samples, well-defined by biomarkers, are needed to evaluate retinal structural and neurovascular changes associated with disease progression in various neurodegenerative diseases.

## Figures and Tables

**Figure 1 jcm-12-01310-f001:**
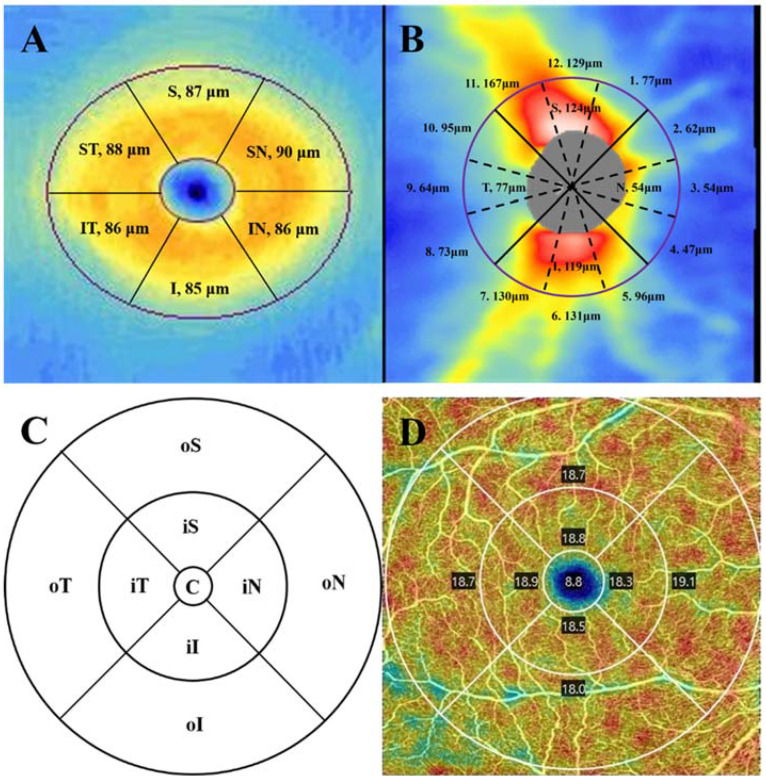
Retinal structural and vascular parameters in OCT and OCTA images of the right eye. (**A**) mGC/IPLT map with six sectors (solid line) in the 14.13 mm^2^ elliptical area. (**B**) cpRNFLT map with four quadrants (solid line) and twelve clock-hour zones (dotted line) in the 3.46 mm-diameter circle scan (purple circle). (**C**,**D**) Diagrams of the macular SCP and the microvascular image. Numeric values in sectors or areas represent average values in the corresponding sectors or areas. Abbreviations: C, center; cpRNFLT, circumpapillary retinal nerve fiber layer thickness; FAZ, foveal avascular zone; i, inner; I, inferior; mGC/IPLT, macular ganglion cell/inner plexiform layer thickness; N, nasal; o, outer; OCT/A, optical coherence tomography/angiography; S, superior; SCP, superficial capillary plexus; T, temporal.

**Figure 2 jcm-12-01310-f002:**
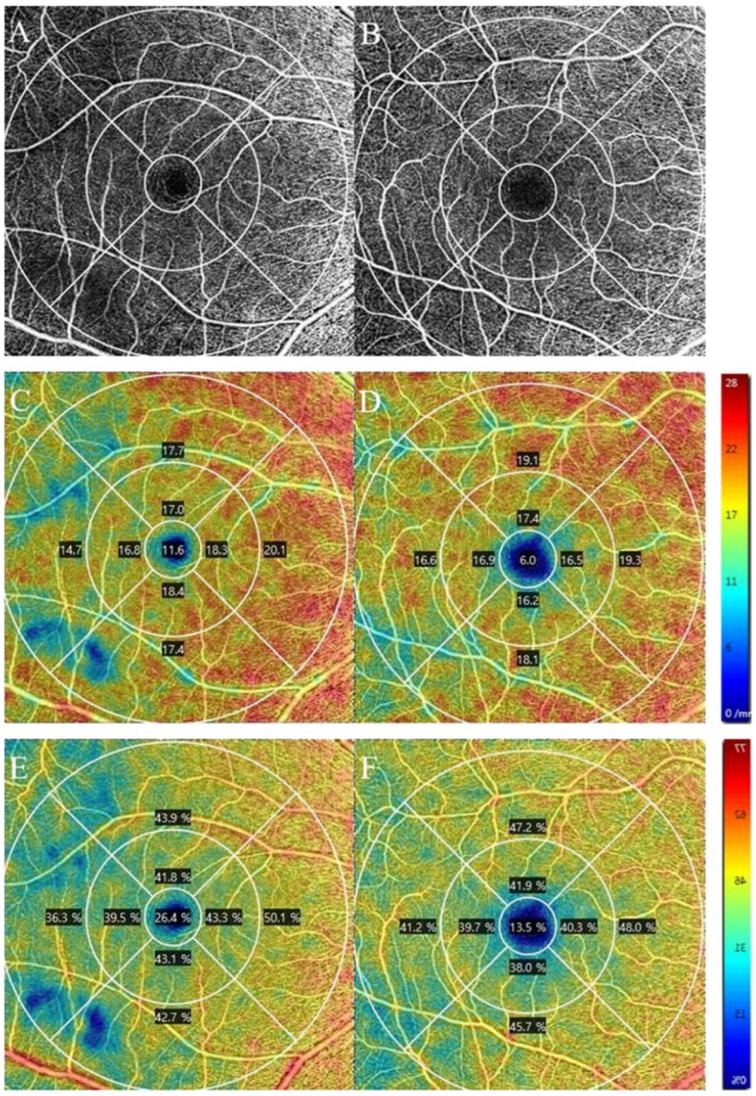
Representative OCTA angiography 6 × 6 mm images of the superficial capillary plexus (SCP) of the right eye from Aβ-positive (A+) MCI and Aβ-negative (A−) MCI participants. En face OCTA of the A+ MCI shows thinner capillaries in the SCP in the temporal macula than A− MCI (**A**,**B**). Color maps for the corresponding quantitative vessel density (**C**,**D**) and perfusion density (**E**,**F**) showed lower microcirculation of temporal macula in the A+ MCI group than in the A− MCI group. Abbreviations: Aβ, beta-amyloid; MCI, mild cognitive impairment; MMSE, Mini-Mental State Examination; OCTA, optical coherence tomography angiography.

**Table 1 jcm-12-01310-t001:** Demographics and clinical characteristics.

	Clinical Diagnostic Groups				Subgroups Based on Aβ Pathology		
	Dementia	MCI	CU	*p*-Value	Aβ-Positive (A+)	Aβ-Negative (A−)	*p*-Value
Number of eyes	27	35	9		42	29	
Age	72.96 ± 7.94	70.26 ± 6.59	70.22 ± 7.05	0.272	71.69 ± 7.43	70.69 ± 6.95	0.450
Sex, female (%)	12 (44.4)	19 (54.3)	6 (66.7)	0.480	23 (54.8)	14 (48.3)	0.635
MMSE	19.78 ± 4.30 ^a,b^	24.77 ± 2.82	27.22 ± 1.92	**<0.001**	21.98 ± 4.50	24.93 ± 3.53	**0.004**
CDR-SB	5.41 ± 2.43 ^a,b^	2.04 ± 1.34 ^a^	0.44 ± 0.39	**<0.001**	3.91 ± 2.74	1.98 ± 1.81	**0.003**
CDR	0.98 ± 0.35 ^a,b^	0.53 ± 0.12	0.33 ± 0.25	**<0.001**	0.77 ± 0.39	0.53 ± 0.23	**0.001**
Aβ positivity (%)	22 (81.5) ^a,b^	17 (48.6)	3 (33.3)	**0.008**			
SBP, mmHg	129.52 ± 16.33	127.46 ± 15.53	131.56 ± 22.06	0.818	131.68 ± 17.78	124.52 ± 13.81	0.050
DBP, mmHg	73.19 ± 11.96	71.63 ± 10.47	72.33 ± 9.82	0.867	73.00 ± 11.42	71.31 ± 10.10	0.558
Hypertension (%)	11 (40.7)	13 (37.1)	1 (11.1)	0.258	17 (40.5)	8 (27.6)	0.318
Diabetes mellitus (%)	3 (11.1)	3 (8.6)	2 (8.0)	0.513	3 (7.1)	5 (17.2)	0.258
Dyslipidemia (%)	7 (25.9)	12 (34.3)	5 (55.6)	0.265	15 (35.7)	9 (31.0)	0.800
Eye laterality, right (%)	14 (51.9)	19 (54.3)	6 (66.7)	0.737	21 (50.0)	18 (62.1)	0.315
BCVA (logMAR)	0.14 ± 0.17	0.11 ± 0.13	0.03 ± 0.07	0.110	0.13 ± 0.17	0.08 ± 0.09	0.474
SE, diopter	−0.02 ± 1.76	0.59 ± 1.67	0.47 ± 1.04	0.405	0.17 ± 1.37	0.59 ± 1.97	0.155
IOP, mmHg	13.37 ± 1.36	14.34 ± 2.92	14.67 ± 3.24	0.535	14.00 ± 2.42	14.03 ± 2.68	0.771
Axial length, mm	23.62 ± 0.60	23.43 ± 0.80	23.20 ± 0.96	0.342	23.44 ± 0.66	23.52 ± 0.88	0.493
CCT, µm	528.04 ± 25.88	527.11 ± 26.88	535.89 ± 24.85	0.514	529.86 ± 28.78	526.72 ± 21.84	0.888

Values are presented as mean ± standard deviation or number of eyes (%), unless otherwise indicated. The significant differences among the three clinical diagnostic groups or the two groups based on amyloid pathology are marked in bold. The significant differences for post-hoc analysis among the three clinical diagnostic groups are indicated as follows: ^a^ *p* < 0.0167 versus CU; ^b^ *p* < 0.0167 versus MCI. Aβ, beta amyloid; BCVA, best-corrected visual acuity; CCT, central corneal thickness; CDR, Clinical Dementia Rating scale; CDR-SB, CDR-Sum of Boxes; CU, cognitively unimpaired; DBP, diastolic blood pressure; IOP, intraocular pressure; logMAR, logarithm of the minimum angle of resolution; MCI, mild cognitive impairment; MMSE, Mini-Mental State Examination; SBP, systolic blood pressure; SE, spherical equivalent.

**Table 2 jcm-12-01310-t002:** Comparison of OCT and OCT angiography parameters in three groups.

	Groups			
	Dementia	MCI	CU	*p*-Value
Retinal structural factor				
Superior quadrant cpRNFLT, μm	110.41 ± 15.07 ^a^	115.09 ± 17.22	125.78 ± 12.67	0.019
5 clock-hour cpRNFLT, μm	87.48 ± 20.29 ^a^	92.83 ± 14.11	102.67 ± 14.47	0.032
6 clock-hour cpRNFLT, μm	116.33 ± 25.21 ^a^	128.83 ± 20.75	141.44 ± 21.10	0.019
Retinal vascular factor				
FAZ circularity index	0.63 ± 0.09 ^a^	0.65 ± 0.11 ^a^	0.74 ± 0.07	0.012
Perifoveal Temporal VD, mm/mm^2^	14.56 ± 2.83 ^a^	15.10 ± 3.10	17.02 ± 2.32	0.046
Perifoveal Mean VD, mm/mm^2^	16.20 ± 2.21 ^a^	16.67 ± 1.76	17.67 ± 1.94	0.040
Perifoveal Superior PD, %	0.40 ± 0.07 ^a^	0.41 ± 0.06	0.45 ± 0.05	0.048
Perifoveal Temporal PD, %	0.36 ± 0.07 ^a^	0.37 ± 0.08	0.42 ± 0.06	0.046
Perifoveal Mean PD, %	0.40 ± 0.06 ^a^	0.41 ± 0.04	0.44 ± 0.05	0.027

Values are presented as mean ± standard deviation. The significantly different parameters among the three clinical diagnostic groups are presented. The significant differences for post-hoc analysis among the three clinical diagnostic groups are indicated as follows: ^a^ *p* < 0.0167 versus CU. cpRNFLT, circumpapillary retinal nerve fiber layer thickness; CU, cognitively unimpaired; FAZ, foveal avascular zone; MCI, mild cognitive impairment; mGC/IPLT, macular ganglion cell/inner plexiform layer thickness; OCT/A, optical coherence tomography/angiography; PD, perfusion density; VD, vessel density.

**Table 3 jcm-12-01310-t003:** Diagnostic performance of OCT and OCTA parameters discriminating patients with dementia and/or MCI from CU controls.

	Cut-Off	AUC	Sensitivity	Specificity
Dementia and MCI (*n* = 62 eyes) versus CU (*n* = 9 eyes)				
Superior quadrant cpRNFLT, μm	123.00	0.772	0.758	0.778
5 clock-hour cpRNFLT, μm	105.00	0.723	0.871	0.556
FAZ circularity index	0.68	0.794	0.661	0.889
Perifoveal Superior VD, mm/mm^2^	17.00	0.717	0.516	0.889
Perifoveal Temporal VD, mm/mm^2^	17.88	0.737	0.903	0.667
Perifoveal Inferior VD, mm/mm^2^	17.77	0.704	0.645	0.778
Perifoveal Mean VD, mm/mm^2^	18.15	0.747	0.823	0.667
Perifoveal Superior PD, %	0.47	0.738	0.984	0.556
Perifoveal Temporal PD, %	0.44	0.735	0.887	0.667
Perifoveal Inferior PD, %	0.45	0.722	0.742	0.778
Perifoveal Mean PD, %	0.45	0.765	0.871	0.667
Total mean VD, mm/mm^2^	17.37	0.729	0.726	0.778
Total mean PD, %	0.44	0.742	0.887	0.667
Dementia (*n* = 27 eyes) versus CU (*n* = 9 eyes)				
Superior quadrant cpRNFLT, μm	118.00	0.821	0.667	0.889
5 clock-hour cpRNFLT, μm	82.00	0.774	0.444	1.000
6 clock-hour cpRNFLT, μm	114.00	0.782	0.481	1.000
FAZ circularity index	0.68	0.844	0.778	0.889
Perifoveal Temporal VD, mm/mm^2^	17.76	0.761	0.926	0.667
Perifoveal Mean VD, mm/mm^2^	17.40	0.757	0.741	0.778
Perifoveal Superior PD, %	0.46	0.753	0.963	0.556
Perifoveal Temporal PD, %	0.44	0.761	0.926	0.667
Perifoveal Mean PD, %	0.43	0.770	0.778	0.778
MCI (*n* = 35 eyes) versus CU (*n* = 9 eyes)				
FAZ circularity index	0.71	0.756	0.714	0.778

AUC, area under the curve; cpRNFLT, circumpapillary retinal nerve fiber layer thickness; CU, cognitively unimpaired; FAZ, foveal avascular zone; MCI, mild cognitive impairment; OCT/A, optical coherence tomography/angiography; PD, perfusion density; VD, vessel density.

**Table 4 jcm-12-01310-t004:** Comparison of OCT and OCTA parameters according to Aβ positivity.

	Aβ-Positive (A+)	Aβ-Negative (A−)	*p*-Value	*p*-Value *
Entire group				
Parafoveal Temporal VD, mm/mm^2^	15.95 ± 2.37	17.04 ± 2.06	0.029	**0.032**
Parafoveal Inferior VD, mm/mm^2^	16.76 ± 1.97	17.33 ± 2.49	0.011	0.285
Parafoveal Mean VD, mm/mm^2^	16.26 ± 2.03	17.05 ± 2.31	0.019	0.089
Parafoveal Temporal PD, %	0.38 ± 0.06	0.41 ± 0.06	0.009	**0.013**
Parafoveal Inferior PD, %	0.40 ± 0.05	0.41 ± 0.07	0.024	0.315
Parafoveal Mean PD, %	0.39 ± 0.05	0.41 ± 0.06	0.016	0.080
Perifoveal Temporal VD, mm/mm^2^	14.43 ± 3.02	16.15 ± 2.63	0.007	**0.031**
Perifoveal Temporal PD, %	0.35 ± 0.08	0.40 ± 0.07	0.009	**0.034**
Total mean PD, %	0.39 ± 0.05	0.41 ± 0.05	0.045	0.185
MCI				
9 clock-hour cpRNFLT, μm	58.06 ± 7.95	51.67 ± 8.02	0.020	**0.029**
Temporal quadrant cpRNFLT, μm	70.71 ± 8.32	64.44 ± 11.45	0.045	0.074
Parafoveal Temporal VD, mm/mm^2^	15.05 ± 2.91	16.97 ± 2.14	0.032	**0.043**
Parafoveal Temporal PD, %	0.35 ± 0.07	0.41 ± 0.06	0.014	**0.022**
Perifoveal Temporal VD, mm/mm^2^	13.87 ± 3.56	16.25 ± 2.10	0.022	**0.022**
Perifoveal Temporal PD, %	0.34 ± 0.09	0.40 ± 0.05	0.041	**0.024**
CU				
Average mGC/IPLT, μm	74.67 ± 2.08	83.17 ± 8.01	0.048	0.200
Superotemporal mGC/IPLT, μm	72.33 ± 4.73	82.00 ± 7.56	0.024	0.305
Inferior mGC/IPLT, μm	73.67 ± 1.53	83.00 ± 7.51	0.024	0.128
Inferotemporal mGC/IPLT, μm	75.00 ± 2.65	84.67 ± 7.17	0.024	0.143
FAZ circularity index	0.66 ± 0.07	0.77 ± 0.04	0.048	0.255

Values are presented as mean ± standard deviation. * Adjusted for MMSE. The significant differences between the two groups based on amyloid pathology are marked in bold. Aβ, beta amyloid; cpRNFLT, circumpapillary retinal nerve fiber layer thickness; CU, cognitively unimpaired; FAZ, foveal avascular zone; MCI, mild cognitive impairment; mGC/IPLT, macular ganglion cell/inner plexiform layer thickness; OCT/A, optical coherence tomography/angiography; PD, perfusion density; VD, vessel density.

## Data Availability

The data presented in this study are available upon request from the corresponding author. The data are not publicly available because informed consent did not include public access to the data used in the study.

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
