# Peer review of "Comparison of Retinal Structural and Neurovascular Changes between Patients with and without Amyloid Pathology"

_jcm, 2023, doi:10.3390/jcm12041310_

Round 1

Reviewer 1 Report (Previous Reviewer 2)

I thank the authors for their changes on the manuscript. A follow up with this change: 

"According to the reviewer’s suggestion, we have reanalyzed the data after including only one eye from each patient, and these analyses reproduced the overall pattern of results. Therefore, we have revised the method and results based on the reanalysis. In addition, to increase statistical power and draw firm conclusions, we have reduced the number of OCT/OCTA variables with excluding lesser important parameters to the theme of our study (e.g., OCTA variables related to the optic disc) in the analysis.

[One eye of each participant was included in the analysis. In cases where both eyes met the eligible criteria, the eye with the worse mean values of structural and vascular parameters on OCT/OCTA images was selected. When the mean values of the parameters were similar in both eyes, the eye with better quality OCT/OCTA image was included.]"

Please explain why one could use date from only one eye. And then when using one eye, why to use the eye with worse value, rather than the other eye or an average of the two eyes. 

A more appropriate approach to address the dependence issues with eyes from the same participant is multi-level modeling. 

Author Response

Multi-level modeling could not be applied to our data, because the OCT/OCTA variables were not normally distributed. Therefore, we reanalyzed the data after including only one eye from each patient. The aim of our study is to evaluate whether retinal structures with microvasculature are associated with underlying beta-amyloid (Aβ) pathologies in patients with Alzheimer’s disease dementia (ADD) and mild cognitive impairment (MCI). Most previous studies have shown reduction of cpRNFLT, mGC/IPLT and retinal vascular density in patients with ADD and MCI, meaning neurodegeneration or underlying neuropathology may affect the retinal structure and vessel density. Based on these considerations, we speculated that the eye with worse value might be more affected one from neurodegenerative conditions, thus using the one might be a more proper way to demonstrate the association between underlying pathology and the related changes of OCT/OCTA variables. Indeed, a few previous studies have examined only worse eye with similar reason (Nouri-Mahdavi et al., Am J Ophthalmol. 2021 Jun;226:172-181). We have cited this previous report in the revised text.    

Reviewer 2 Report (New Reviewer)

The authors recruited twenty- seven patients with dementia, 35 with MCI, and 9 with cognitively unimpaired (CU) controls. 

One eye of each participant was included in the analysis. Based on amyloid PET or CSF Aβ level results, all participants were subdivided into A+ (n = 42 eyes) or A− (n = 29eyes) groups.

Using optical coherence tomography (OCT) and OCT angiography, retinal circumpapillary retinal nerve fiber layer thickness (cpRNFLT), macular ganglion cell/inner plexiform layer thickness (mGC/IPLT), and microcirculation of the macular superficial capillary plexus 

and radial peripapillary capillaries were measured.

The major finding of this study is 

1.structural and vascular parameters did not differ between the A+ and A– with dementia.

2.structural and vascular factors significantly decreased in the order of:  dementia < MCI < CU controls.

3.cpRNFLT was greater in the A+ than in the A– with MCI

4.mGC/IPLT was lower in the A+ CU than in the A– CU. 

I would like the authors to clarify the following:

Q1 "cpRNFLT was greater in the A+ than in the A– with MCI" "structural and vascular parameters did not differ between the A+ and A– with dementia." Is there difference of cpRNFLT between the A+ and A– with CU?

Q2 Interpretation to imply that contributions of Aβ pathology to retinal degeneration and microvascular changes begin at the preclinical stage (CU group) and play a role in the early and mild stages of disease (MCI group), but may not be emphasized in the dementia stage . This perspective of data interpretation may answer to the previous discrepancies among studies on interaction between AB and retinal vascular factors within each clinical diagnostic group. The authors explained that " retinal structural changes have been reported in other neurodegenerative diseases, such as frontotemporal dementia, Parkinson’s disease, and Huntington’s disease". I was wondering if the author perform analysis comparing retinal structural and vascular changes of A- Dementia with A+MCI and/or A+CU. Q3 "increased 9-clock-hour and temporal cpRNFLT in the A+ MCI group may be explained by a paradoxical increase in RNFLT in the MCI stage attributed to gliosis occurring prior to neuronal loss."  Different clock-hour cpRNFLT show distinct relationship with other parameters. e.g. 5 and 6clock-hour cpRNFLTs were significantly different between the three groups, with generally smaller thickness in the direction of dementia < MCI < CU controls.    In the MCI group, the 9 clock-hour cpRNFLT values was significantly increased in the A+ group than in the A− group.   3 clock-hour cpRNFLT was positively correlated with MMSE scores in the entire A− group.   Please discuss why paradoxical increase in RNFLT in the MCI stage only in 9 clock-hour cpRNFLT values.

Author Response

Q1 "cpRNFLT was greater in the A+ than in the A– with MCI" "structural and vascular parameters did not differ between the A+ and A– with dementia." Is there difference of cpRNFLT between the A+ and A– with CU?

Response: As we noted in the supplementary Table S4, there was no difference of cpRNFLT between the A+ and A- with CU. However, it should be cautiously interpreted, because the numbers of A+ (N=3) and A- CU (N=6) groups were too small. 

Q2 Interpretation to imply that contributions of Aβ pathology to retinal degeneration and microvascular changes begin at the preclinical stage (CU group) and play a role in the early and mild stages of disease (MCI group), but may not be emphasized in the dementia stage . This perspective of data interpretation may answer to the previous discrepancies among studies on interaction between AB and retinal vascular factors within each clinical diagnostic group. The authors explained that " retinal structural changes have been reported in other neurodegenerative diseases, such as frontotemporal dementia, Parkinson’s disease, and Huntington’s disease". I was wondering if the author perform analysis comparing retinal structural and vascular changes of A- Dementia with A+MCI and/or A+CU.

Response: As the reviewer recommended, we performed analysis comparing retinal structural and vascular changes of A- Dementia with A+MCI and/or A+CU (Kruskal–Wallis test, please see below the table). As we expected based on our hypothesis and the previous studies about OCT changes of other neurodegenerative diseases (FTD, IPD, HD), A-dementia group showed more reduced cpRNFLT, mGC/IPLT than A+MCI, but not A+CU. However, because the numbers of A-Dementia group (n=5) and A+CU group (n=3) were too small, longitudinal studies with larger samples, well defined by biomarkers, are needed to evaluate retinal structural and neurovascular changes associated with disease progression in various neurodegenerative diseases, as we noted in the first revised discussion.

A− Dementia (n=5)

A+ MCI (n=17)

A+ CU (n=3)

P-value

Retinal structural factor

Average mGC/IPLT, μm

77.20 ± 5.31

81.35 ± 4.31

74.67 ± 2.08

0.031

Superior mGC/IPLT, μm

76.00 ± 6.60

81.71 ± 4.66

74.33 ± 4.93

0.042

Inferior mGC/IPLT, μm

74.40 ± 4.45

78.94 ± 5.10

73.67 ± 1.53

0.041

Inferotemporal mGC/IPLT, μm

77.60 ± 4.34

81.88 ± 4.05a

75.00 ± 2.65

0.021

Inferior quadrant cpRNFLT, μm

99.40 ± 14.52

117.41 ± 10.30

124.67 ± 17.47

0.034

Q3 "increased 9-clock-hour and temporal cpRNFLT in the A+ MCI group may be explained by a paradoxical increase in RNFLT in the MCI stage attributed to gliosis occurring prior to neuronal loss."  Different clock-hour cpRNFLT show distinct relationship with other parameters. e.g. 5 and 6 clock-hour cpRNFLTs were significantly different between the three groups, with generally smaller thickness in the direction of dementia < MCI < CU controls. In the MCI group, the 9 clock-hour cpRNFLT values was significantly increased in the A+ group than in the A− group. 3 clock-hour cpRNFLT was positively correlated with MMSE scores in the entire A− group. Please discuss why paradoxical increase in RNFLT in the MCI stage only in 9 clock-hour cpRNFLT values.

Response: Approximately 50% of the retinal ganglion cells are located within 4.5 mm (16º) of the fovea center, a macula region that comprises only 7.3% of the total area. (J Comp Neurol. 1990 Oct 1;300(1):5-25.). In addition, all axon of retinal ganglion cells from the superior macula and some axons of the inferior macula are projected to the temporal quadrant of the disc (Prog Retin Eye Res. 2013 Jan;32:1-21.). Therefore, contribution of Aβ pathology to retinal degeneration, gliosis or inflammatory reaction which can cause paradoxical increase in RNFL thickness might be more prominent at the temporal quadrant of the disc (9 clock-hour) in the early/mild stage of the disease. We have added this discussion in the revised manuscript.

This manuscript is a resubmission of an earlier submission. The following is a list of the peer review reports and author responses from that submission.

Round 1

Reviewer 1 Report

The authors propose a cross-sectional comparison of retinal structure and microvasculature in 27 cases with dementia, 35 MCI cases and 9 controls. They used various retinal parameters such as mCG-IPLT, cpRNFLT (retinal structure) and FAZ circularity, perifoveal temporal and mean VD and PD as well as optic disc inferior PD. Importantly, they performed an PET or CSF amyloid characterization of all of the cases in order to explore the impact of amyloid pathology on retinal structural and vascular changes. All retinal parameters were assessed using OCT and OCTA imaging. As expected, some among the retinal structural and vascular factors showed marked decreases in the dementia group (inferonasal mCG-IPLT, superior quadrant cpRNFLT, all retinal vascular factors). The proposed ROC curves for dementia-MCI versus controls distinction are of modest performance with AUC < 0.8. When amyloid positivity was considered, OCT and OCTA parameters differed in MCI and CU cases (although barely significant) but not in AD cases. ROC curves remained here again of low performance. Most of the parameters studied are positively related to MMSE scores (with the exception of parafoveal superior PD in the A+ group.

The significance of retinal changes in AD (mainly in prodromal and preclinical cases) are highly disputed and exhaustively studied.  They have been used as promising candidate biomarkers of MCI conversion to AD (for metaanalysis Ge et al., Ageing Res Rev 2021;69). OCT and OCTA biomarkers have been also extensively studied with more than 70 studies in this field with mixed results (Song et al., Eye Brain 2021;13).  The present data is a valuable effort to examine the utility of these markers when considering amyloid positivity. However, there are serious methodological problems that decrease my enthusiasm for this contribution:

Multiple comparison corrections were used only for demographic and clinical data but not in Table 2 and 4 (despite an impressive number of comparisons). A Benjamini-Hochberg correction is needed here and would probably lead to the suppression of most of the significant data.

The question of the collinearity between the OCT/OCTA variables is not addressed, yet it can decisively impact on the observed associations. The authors examine the correlation matrices for MMSE but not for the OCT/OCTA variables.

Most importantly, what is critically missing here is the research question. If the main issue is to predict the distinction between MCI/controls, the authors should build regression models with OCT/OCTA variables (after testing for collinearity), Aß positivity, and MMSE scores (independent predictors). The same approach should be used for the dementia/control distinction (taking into account that the former group is heterogeneous and poorly characterized).

Using ROC curves makes no sense here before clearly defining a few variables using regression models that resist to multiple comparison corrections.

Reviewer 2 Report

The main concern that I have with the paper lies in the data analysis. As some eyes are from the same participant, not all the cases (here the eyes) are independent from each other to qualify for the statistical analyses (ANOVA, chi-square, etc). The dependences between some pairs of eyes should be considered before the results could be interpreted.

Minor: what does NIA-AA stand for?

It is not clear whether the eyes are post-mortem or not?

What do “consecutive eyes” mean?